# Is large improvement in efficiency of impulsive noise removal in color images still possible?

**Lukasz Malinski**[1]*, **Krystian Radlak**[2,3], **Bogdan Smolka**[2]

**1** Division of Industrial Informatics, Silesian University of Technology, Katowice, Poland, **2** Department of Data Science and Engineering, Silesian University of Technology, Gliwice, Poland, **3** Institute of Computer Science, Warsaw University of Technology, Warsaw, Poland

* E-mail: lukasz.malinski@polsl.pl

**Data Availability Statement:** All original images are avaliable from: https://www.kaggle.com/lmalinski/training-image-set. All processed images are avaliable from: https://www.kaggle.com/lmalinski/impulsive-noise-removal.

## Abstract

The substantial improvement in the efficiency of switching filters, intended for the removal of impulsive noise within color images is described. Numerous noisy pixel detection and replacement techniques are evaluated, where the filtering performance for color images and subsequent results are assessed using statistical reasoning. Denoising efficiency for the applied detection and interpolation techniques are assessed when the location of corrupted pixels are identified by noisy pixel detection algorithms and also in the scenario when they are already known. The results show that improvement in objective quality measures can be achieved by using more robust detection techniques, combined with novel methods of corrupted pixel restoration. A significant increase in the image denoising performance is achieved for both pixel detection and interpolation, surpassing current filtering methods especially via the application of a convolutional network. The interpolation techniques used in the image inpainting methods also significantly increased the efficiency of impulsive noise removal.

## Introduction

Different types of impulsive noise decrease the quality of digital, color images and may be caused through transmission errors, electromagnetic disturbances, ageing of the storage material, sensors imperfections, and flawed memory regions [1–4]. Impulsive noise can be also introduced to images deliberately, due to the fact that deep neural networks are vulnerable to adversarial examples [5]. For instance, in one-pixel attacks, altering only few pixels in an original image is enough to fool a deep neural network [6, 7]. Therefore, filtering algorithms dedicated to the suppression of impulsive disturbances in color images and also considered as defensive methods against adversarial attacks, have attracted considerable interest among many researchers [8–17].

In recent years, intensive development of the methods used for noise reduction in digital images has been observed [18]. More advanced methods have indeed been proposed. However, doubts have appeared as to whether these new methods are able to significantly increase

**Funding:** National Science Centre, Poland (Narodowe Centrum Nauki (PL)) Award number: 2017/25/B/ST6/02219 Receiver: Bogdan Smolka.

**Competing interests:** The authors have declared that no competing interests exist.

the objective indicators associated with denoising quality [19, 20]. In the case of Gaussian noise, progress by the standard quality measures is tending to stagnate, it is therefore assumed that the new methods will no longer be able to significantly ameliorate the standard indicators of image quality. Numerous studies suggest that current methods are already quite close to the theoretical, impassable, restoration quality limit [21]. The situation is different in the case of impulsive noise removal. Most of the modern methods use switching filtering, which first detects noise induced image corruption and the noisy pixels are then replaced by estimates, calculated from non-corrupted pixels in their local neighborhood. These methods therefore focus on the problem of detecting damaged pixels and less importance is given to how they are replaced. The focus of this work is to examine the effective interpolation techniques and assess where the application of these methods enables significant improvement in denoising efficiency.

Generally, algorithms designed for impulsive noise reduction are based on contextual processing schemes which replace corrupted pixels with their estimates, using information obtained from the local sliding processing window. The most popular algorithm developed to cope with impulsive noise is the Vector Median Filter (VMF) [22] in which the main drawback is processing of every pixel in the image, regardless of whether it is noisy or not. Therefore, various switching filters were designed to repair only those pixels found to be corrupted by the impulsive noise process [23–29]. Among them, methods based on: reduced vector ordering [30–32], peer group concept [33–35], quaternions [36–38] and fuzzy sets theory [39–42], have achieved satisfying results.

Another family of efficient image denoising algorithms utilizes a sparse linear combination of basis elements taken from a learnt dictionary adapted to the processed data [43–45]. Initially it was applied to attenuate the Gaussian noise, however soon the method was extended so that the mixture of Gaussian and heavy tailed noise as well as imaging artifacts could also be attenuated [46–50]. This technique is also capable of suppressing the impulsive noise [51–55].

It can be observed that most of the efforts focused on the development of switching techniques have been centered on impulse detection, while VMF was used as an estimator for the detected corrupted pixels [35, 56, 57]. The VMF computes in the first step, the sum of distances from each pixel to all samples in the processing window. Then the central pixel is replaced by the one for which the sum of distances is minimized. When the corrupted pixels are detected, they can be replaced using the Arithmetic Mean Filter (AMF), operating only on uncontaminated pixels [35, 57–59]. The denoising performance of such a trimmed AMF marginally improves when using VMF and it is significantly less computationally demanding. Both estimators are resistant to the occurrence of local outliers, but although AMF is much faster, VMF is useful when every pixel in the local neighborhood is classified as noisy. Therefore, the combined approach has been introduced in [58], which computes the trimmed mean when the neighborhood contains undistorted pixels and VMF otherwise. This algorithm is very reliable, fast and performs competitively with respect to the state-of-the-art filters [57–59], therefore this interpolation technique was chosen as *Reference Estimator* (RE) for the experiments reported in this work.

In recent years, there has been a lack of significant improvement in the performance of the switching filters. Many of the recently introduced filters achieve comparable results, when evaluated with benchmark images. It can be attributed to the main focus of research efforts being mainly towards impulse detection, where the accuracy is already very close to the current theoretical maximum. Further improvement in the precision of impulse detection is very challenging and might only be achieved with very complex and computationally expensive solutions. Therefore, in our investigation whether a substantial increase in the efficiency of impulsive noise suppression is still possible, our focus is on the switching filters estimation step.

For this purpose, data interpolation techniques are described, which to our knowledge have not yet been utilised for impulse removal in color images. Among those, very promising results are provided by the methods used in image inpainting [60]. Additionally, the performance of neural networks designed for detection and restoration of impulses was also examined [61, 62]. A similar approach, to find alternative methods for impulse noise reduction have been previously utilized [63–67].

In following Section the concept of switching filtering is presented, with explicit division into impulse detection and replacement stages. Farther in the paper, simulation performance tests are described demonstrating significant denoising improvement that can be achieved on both detection and impulse replacement parts of the investigated filters. In addition, the impact of imperfections in impulse detection is considered, followed by visual comparison of image enhancement results. Finally some conclusions are drawn on the merits of the investigated filtering designs.

## Switching algorithms

A switching filter is composed of two stages: impulse detection and its replacement, using an appropriate estimation technique. For both processing steps, a number of different approaches have been proposed in the extensive literature [27, 68–71], and a selected subset of those algorithms is briefly described in the following subsections.

The concept of a switching filter is presented in Fig 1. The impulse detection step is responsible for assigning pixels of the processed image into two classes: noisy and clean. Pixels are then replaced by some particular algorithmic methods, which try to estimate the original color, using information obtained from the local neighborhood. Pixels which are classified as clean, are not altered in any way.

## Impulse detection

In general, impulse detection is responsible for controlling the data flow in the switching filter. Its input is comprised of both the corrupted image and some additional parameters, the most common among which is the size of the operating window, where others are algorithm specific.

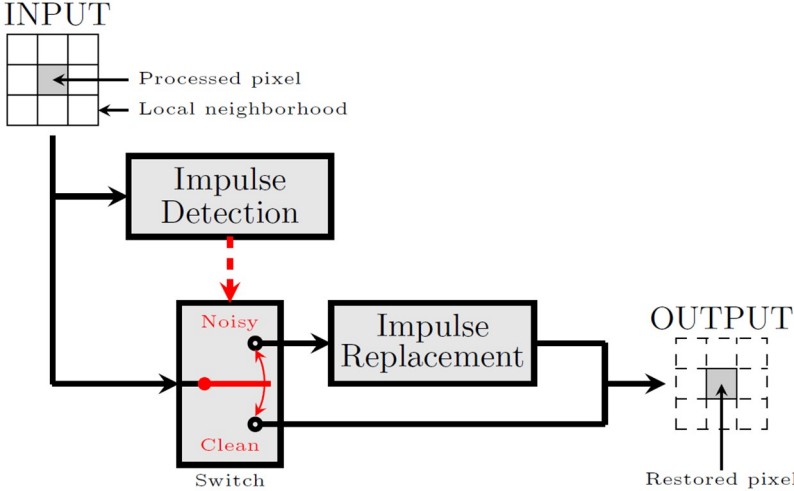

**Fig 1. Switching filter for impulsive noise removal.**

The minimum operating window size and the filtering parameters recommended by the authors of these filtering schemes are used in each of the detection techniques evaluated in this paper. The output of an impulse detection step is a noise map, which is a binary array of: *zeros*, depicting the detected noisy pixels, and *ones* marking those classified as clean. In this paper, impulse detection algorithms will be often referred to as *Detectors*, and all abbreviations of detection schemes names will by marked with D as the last letter of each corresponding acronym. The detectors investigated in this work are enumerated in Table 1(a)).

## Impulse replacement

After noisy pixels are recognized in the switching filter detection step, the impulsive pixel restoration is performed to replace corrupted pixels with their estimates, obtained using information derived from the local neighborhood. In this paper, such impulse suppression algorithms are called *Estimators*, and all abbreviations of their names contain E as the last letter of each corresponding acronym.

From a plethora of available algorithms, a representative subset of methods, which are characterized with both competitive impulse suppression performance and reasonable computational cost, were selected and presented in Table 1(b)).

It is important to mention that the original implementation of algorithms and pre-trained neural networks are prepared for different computational platforms (MATLAB, Python and C++). Also some parts of the code are only provided in binary form, therefore detailed comparison of complexity between those computational implementations is not feasible and of little importance to this work. Moreover, due to multi-platform problem, the computations for different methods are performed on different machines mostly using CPU but also GPU in case of neural networks. Therefore, measurements of the computational time are incomparable and omitted from this paper.

## Performance tests

All of the algorithms (detectors and estimators) selected for evaluation have been tuned using parameters recommended by the authors of the respective papers. Also, our efforts focused on determining the efficiency of these estimators when locations of impulses are known and when they are found by a detection algorithm.

For each of the tests an image dataset was chosen (available for download from [88]), composed of 100 color images with a resolution of $640 \times 480$. Images are artificially contaminated by Random Valued Impulsive Noise (RVIN) for the following densities: $\rho \in \langle 0.1, 0.3, 0.5 \rangle$. One of the most common, the RVIN model was used—Channel Together Random Impulse [89]. In this model only a given fraction $\rho$ of the image pixels is affected by the noise process. If a particular pixel is corrupted, then all of its RGB channels are randomly altered by a value taken from full 8-bit RGB encoding range: $\langle 0, 255 \rangle$.

Next, the integrated detectors and estimators described in previous Section are paired together. This way every detector is matched with an estimator to form a switching filter. Also, in order to eliminate any detection imperfections and to prevent impulse suppression efficiency decrease, the true maps of noise (acquired during image contamination) are given, which are considered as an output of the Perfect Detector (PD). Finally, for each contaminated image:

- The evaluated accuracy (ACC) and F1-score of each detector is given.

- The denoising efficiency of each detector and estimator pair is assessed.

**Table 1. Description of the considered impulsive noise *Detectors* and *Estimators*.**

| DETECTOR | Full name of the detector | |
|---|---|---|
| ACWD | Adaptive Central-Weighed Vector Median Filter | [72] |

computes the dissimilarity between the central pixel of the processing window $W$ and the outputs of the Adaptive Central-Weighed Vector Median Filter (ACWVMF) [72] obtained for increasing weighting coefficients. If the sum of Euclidean distances between pixels of $W$ determined for a set of weighting factors exceeds a predefined threshold, then the processed pixel is classified as noisy.

| | | |
|---|---|---|
| IDCNN | Impulse Detection Convolutional Neural Network | [73] |

is a modification of the Denoising Convolutional Neural Network (DnCNN) [74]. IDCNN consists of a sequence of convolutional layers followed by Rectified Linear Unit [75] and Batch Normalization [76] for feature extraction and sigmoid layer for the noisy pixels detection. Pretrained model recommended in [73], trained with default parameters summarized in Table 2 was used.

| | | |
|---|---|---|
| FFD | Fast Fuzzy Noise Reduction Filter | [77] |

evaluates a similarity of pixels in the neighborhood using a fuzzy metric and is applied for impulse detection in Fast Fuzzy Noise Reduction Filter [77]. If the central pixel of the filtering window is not the most similar to its neighbors, then it is assumed to be noisy.

| | | |
|---|---|---|
| FASTD | Fast Adaptive Switching Trimmed Arithmetic Mean Filter | [57, 58, 78] |

computes the pixel impulsiveness measure, based on the trimmed sum of ordered Euclidean distances in the RGB color space. This measure is then adapted to local image characteristic and compared to a predefined threshold. If the threshold value is exceeded, the pixel is classified as corrupted.

| | | |
|---|---|---|
| FPGD | Impulse detection algorithm based on the Fast Peer-Group Filter | [34, 35] |

utilizes a simplified peer-group based classification technique. It categorizes neighbors of the processed pixel into two groups: peers and non-related pixels, judging upon their Euclidean distance in the RGB color space. Finally, if the processed pixel has less than two peers in $W$, it is classified as noisy.

| | | |
|---|---|---|
| GDPD | Geodesic Digital Path Detector | [79, 80] |

determines the minimal connection cost between the border of $W$ and its center, which is the pixel being processed. If the connection cost of a minimum path exceeds a predefined threshold, then it indicates that the central pixel is corrupted, as no low-cost connection with the window boundary can be found.

(a)

| ESTIMATOR | Full name of the estimator | |
|---|---|---|
| VSIE | Very Straightforward Interpolation (replacement) of NAN's | [81] |

performs a one-dimensional data interpolation based on uncorrupted values in a dataset. Each color image channel is processed as an individual 2D-array, where the column-wise and row-wise estimates of noisy pixels are averaged. For the experiments described in this paper, a shape-preserving piecewise cubic interpolation method was adopted.

| | | |
|---|---|---|
| FBE | Fill Bad | [82] |

estimator computes missing values in the image channel array using a common linear interpolation which estimates a new channel value using linear weighted averaging. The algorithm exploits only the values of the adjacent neighbors of a corrupted pixel, processing the image channel column-wise only.

| | | |
|---|---|---|
| DCTE | Discrete Cosine Transform | [83] |

based estimator was created for automatic smoothing of multidimensional incomplete data and adopted for the purpose of gap reconstruction within large datasets, such as medical or satellite images. The algorithm is based on penalized least square method and is processing image channels by performing DCT and Inverse DCT for some desired number of iterations. The amount of smoothing is automatically adjusted applying minimization of generalized cross validation score.

| | | |
|---|---|---|
| IPNE | InPaint NANs | [84] |

calculates missing data by establishing sets of linear equations for every missing value in the array (image channel). Each equation is constructed using the assumption that every value in the array is an average of the adjacent ones (4-neighborhood). As the number of equations exceeds the number of missing values, the least squares method is used to compute the final estimates.

| | | |
|---|---|---|
| NSE | Navier-Stokes Estimator | [85] |

estimator is a heuristic inpainting technique based on computational fluid dynamics methodology. It was created as a result of research showing that there is an analogy between the image and the stream function of a two-dimensional incompressible flow of fluid. The aim of the algorithm is to continue isophotes while matching gradient vectors at the boundary of the missing data regions.

| | | |
|---|---|---|
| CNNE | Convolutional Neural Network | [74] |

estimator is based on a similar design like CNND but consists of only seven layers, in which the first layer contains *Dilated Convolution+ReLU*, then next five consecutive layers are *Dilated Convolution+Batch Normalization+ReLU*, and the last layer is *Dilated Convolution*. A set of these CNN denoisers are pre-trained with different noise levels and integrated into the optimization-based framework in order to restore distorted images. This approach is also used for inpainting purposes and in this case, in the first step the missing pixels are interpolated using Shepard interpolation [86] and then the interpolation artifacts are removed using a model-based optimization of an inverse problem.

(b)

**Table 2. Summary of CNND detector parameters [73].**

| Parameter | Value/Method |
|---|---|
| Number of convolutional layers | 17 |
| Number of filters in convolutional layer | 64 |
| Size of convolutional window | $3 \times 3$ |
| Number of epochs | 50 |
| Learning rate | 0.001 |
| Learning rate decay | 0.1 |
| Epoch in which learning rate decay is used | 30 |
| Batch size | 128 |
| Weights optimization | ADAM optimizer [87] |
| Patch size | $41 \times 41$ |

## Impulse detection

In our experiments, the performance of the chosen impulse detection algorithms was evaluated using Accuracy (ACC) and F1-score:

$$\text{ACC} = \frac{|\text{TP}| + |\text{TN}|}{|\text{TP}| + |\text{TN}| + |\text{FP}| + |\text{FN}|}, \qquad \text{F1} - \text{score} = \frac{2 \times \text{R} \times \text{P}}{\text{R} + \text{P}}, \tag{1}$$

where:

$$\text{R} = \frac{|\text{TP}|}{|\text{TP}| + |\text{FN}|}, \quad \text{P} = \frac{|\text{TP}|}{|\text{TP}| + |\text{FP}|}, \tag{2}$$

and |TP|, |TN|, |FP| and |FN| are numbers of pixels assigned to following states:

- True Positive (TP)—correctly detected impulse.

- True Negative (TP)—pixel recognized as noise-free.

- False Positive (FP)—pixel falsely classified as impulse (Type-I error).

- False Negative (FN)—pixel incorrectly classified as noise-free (Type-II error).

For all concerned impulse detection techniques, an F1-score was computed for every image of the dataset [88] (also dataset containing all noisy and restored images is available online at [90]) using noise fractions: $\rho \in \{0.1, 0.3, 0.5\}$. Medians of obtained ACC and F1-score values are summarized in Table 3.

The analysis of the obtained results leads to the following conclusions:

- The most accurate detector, among tested, is without any doubt the CNND, which achieves an astonishingly high F1-score and ACC, even for $\rho = 0.5$.

- FPGD algorithm is not very accurate for low noise ratios but its performance improves substantially if higher noise fraction occurs.

- Detection efficiency of ACWD and FFD, while rather satisfactory for low impulse density, significantly deteriorates when more impulses are present.

- FASTD and GDPD techniques are robust to the increase in impulse occurrence ratio, which makes them the second and third best detectors among tested.

**Table 3. Medians of ACC and F1-score obtained using different detectors.**

| $\rho$ | ACWD | CNND | FASTD | FFD | FPGD | GDPD | ACWD | CNND | FASTD | FFD | FPGD | GDPD |
|---|---|---|---|---|---|---|---|---|---|---|---|---|
| | ACC | | | | | | F1-score | | | | | |
| 0.1 | 0.9950 | **0.9996** | 0.9965 | 0.9945 | 0.9921 | 0.9934 | 0.9750 | **0.9978** | 0.9823 | 0.9728 | 0.9613 | 0.9670 |
| 0.3 | 0.9653 | **0.9990** | 0.9904 | 0.9788 | 0.9831 | 0.9859 | 0.9390 | **0.9983** | 0.9840 | 0.9635 | 0.9719 | 0.9765 |
| 0.5 | 0.8591 | **0.9956** | 0.9707 | 0.9062 | 0.9598 | 0.9714 | 0.8388 | **0.9956** | 0.9706 | 0.8978 | 0.9605 | 0.9715 |

Best results are emboldened.

Box plots shown in Fig 2, present the statistical features: median, quartiles: Q1 and Q3, and whiskers: min/max, of the distribution of F1-score values obtained for every corrupted image in dataset and for each detector.

It can be observed that:

- F1-score medians are explicitly different for every detection technique, especially if noise fraction $\rho$ is high.

- Different algorithms provide significantly distinct distributions of F1-score within a tested dataset of images.

- The CNND strongly outperforms other detectors.

- The performance of ACWD deteriorates significantly along with an increase in noise fraction.

## Impulse replacement

For noise replacement efficiency assessment the Peak Signal-to-Noise Ratio (PSNR) was used:

$$\text{PSNR} = 10 \ \log_{10} \frac{255^2}{\text{MSE}}, \qquad \text{MSE} = \frac{1}{3\theta} \sum_{u=1}^{\mu} \sum_{v=1}^{v} \sum_{q} (o_{u,v}^q - x_{u,v}^q)^2, \tag{3}$$

where $o_{u,v}^q$ and $x_{u,v}^q$, $q \in \{R, G, B\}$ are the channels of the original and restored image pixels, $v, \mu$ denote image sizes and $\theta$ stand for the overall number of image pixels.

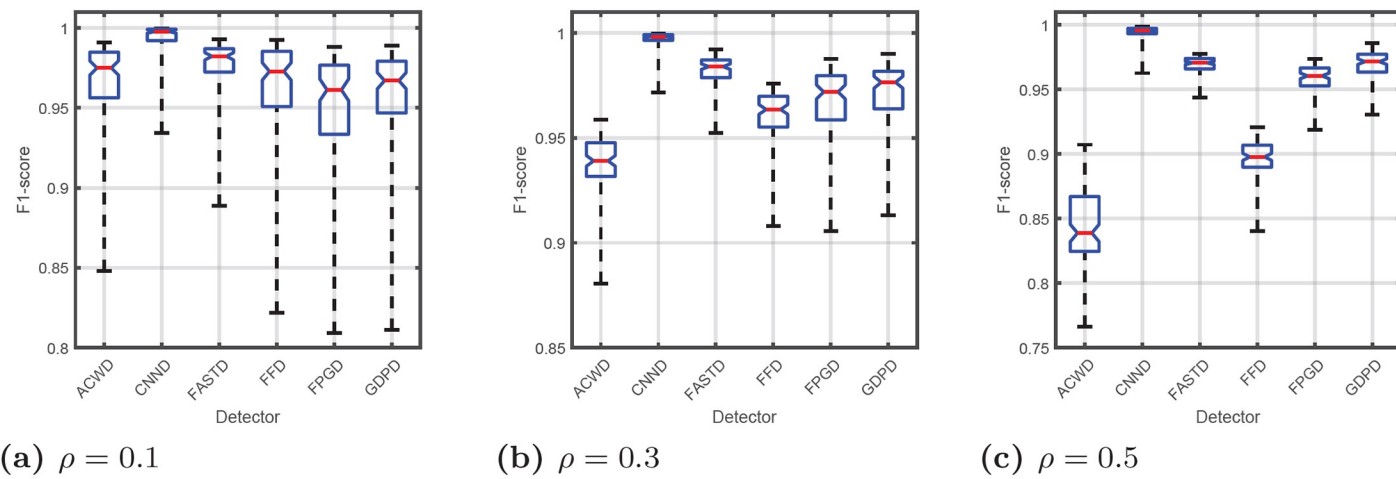

**(a)** $\rho = 0.1$ **(b)** $\rho = 0.3$ **(c)** $\rho = 0.5$

**Fig 2. Impulse detection performance.** Evaluated with F1-score measure.

The aim of this paper is to show a statistically significant improvement in RVIN removal. Also, to make this work as clear and consistent as possible the inclusion of other popular quality measures, such as the Feature Similarity Index (FSIM) [91] or Normalized Color Difference (NCD) [92] is beyond the scope of this paper, especially when statistical tests should be also performed on those additional quality measures. Therefore, for every corrupted image restored using each detector-estimator pair, the PSNR measure only was computed. The medians of PSNR for RE are summarized in the first column of Table 4. Other columns depict differences between the median obtained for particular pair and when computed for a detector paired with RE.

Results can be summarized as follows:

- If PD is used, the overall impulse replacement performance, deteriorates along with an increase in noise fraction, but the relative average efficiency among different algorithms is preserved.

- Also when PD is used, most of tested estimators outperform the RE. However, for highest noise fraction even FBE and NSE algorithms perform worse than the reference one (RE). Generally, FBE performs less efficiently in every case, and NSE shows similar efficiency.

- The most efficient estimator is the CNNE and the second one is IPNE. Also results for DCTE show noticeable improvement and VSIE performs mostly better than RE.

- In the case when CNNE is paired with PD or CNND, the median of PSNR obtained is more than 4 dB higher, which is really spectacular. For other detectors this improvement is not so prominent, but still quite high.

- The adopted detection scheme has a significant impact on the noise suppression performance, especially if noise fraction is high. Generally, the less accurate of a detector being used, superiority of other estimators over the RE is less significant, or not observed at all.

- Even the best estimator (CNNE) may perform worse than reference technique if the detector has low F1-score and noise density is high enough. This suggests that the more sophisticated estimators are less robust to detection imperfections. Also it shows that although the RE is relatively simple and computationally efficient, it can still be regarded as a robust and competitive noise suppression technique.

Also box plots, for estimators paired with best detector (CNND) only, are presented in Fig 3. Those, present the statistical features of PSNR distribution obtained for each estimator together with the results obtained for PD. It can be seen that even though median values differ, the variations of results are not very distinct.

Thorough statistical analysis of the obtained results was achieved, where Friedman's test [93] was applied for a significance level of $\alpha = 0.05$, two hypothesizes are subsequently formulated:

H0:. Differences in the obtained results are not statistically significant ($p \geq 0.05$).

H1:. Results obtained for all algorithms are significantly different ($p < 0.05$).

In every test scenario, the H0 hypothesis was rejected with probability of error $p < 0.001$. This means that performance of particular noise suppression algorithm differs significantly, no matter which detector is used. Table 5 depicts mean ranks computed for Friedman's test. If the mean rank for particular estimator is lower than the mean rank obtained for RE (in the same row), it means that this algorithm is on average less efficient than RE (for the same

Table 4. Medians of PSNR for RE and differences between other estimators and RE, paired with all detectors.

| Detector | Estimator | | | | | | | | | | | | | | | | | | | | |
|---|---|---|---|---|---|---|---|---|---|---|---|---|---|---|---|---|---|---|---|---|---|
| | ρ = 0.1 | | | | | | | ρ = 0.3 | | | | | | | ρ = 0.5 | | | | | | |
| | RE | VSIE | CNNE | FBE | NSE | DCTE | IPNE | RE | VSIE | CNNE | FBE | NSE | DCTE | IPNE | RE | VSIE | CNNE | FBE | NSE | DCTE | IPNE |
| ACWD | 36.19 | 0.98 | **2.55** | 0.02 | 0.00 | 1.32 | 1.38 | 24.63 | -0.07 | **0.10** | -0.56 | -0.08 | -0.39 | -0.48 | 17.24 | -0.17 | -0.09 | -0.51 | -0.08 | -0.50 | -0.60 |
| CNND | 40.11 | 1.60 | **4.18** | -0.78 | -0.28 | 2.32 | 2.56 | 34.39 | 0.98 | **3.70** | -1.34 | -0.13 | 2.02 | 2.08 | 29.84 | -0.12 | **1.93** | -1.67 | -0.13 | 0.83 | 0.72 |
| FASTD | 38.67 | 1.04 | **2.94** | -0.60 | 0.04 | 1.73 | 1.83 | 32.90 | 0.51 | **2.19** | -1.23 | 0.13 | 0.88 | 0.92 | 26.02 | 0.02 | **0.81** | -1.12 | -0.80 | -0.75 | -0.97 |
| FFD | 36.84 | 0.89 | **2.75** | -0.12 | -0.27 | 1.13 | 1.31 | 28.43 | 0.04 | **0.44** | -0.62 | -0.10 | -0.37 | -0.47 | 19.62 | -0.20 | -0.03 | -0.66 | -0.18 | -0.61 | -0.74 |
| FPGD | 36.75 | 0.99 | **2.75** | -0.23 | 0.26 | 1.61 | 1.71 | 31.91 | 0.56 | **2.55** | -1.16 | 0.06 | 1.16 | 1.31 | 28.02 | 0.04 | **1.79** | -1.40 | -0.41 | 0.58 | 0.53 |
| GDPD | 37.05 | 0.80 | **2.37** | -0.39 | 0.15 | 1.32 | 1.52 | 32.16 | 0.54 | **2.31** | -1.08 | 0.15 | 1.08 | 1.16 | 28.42 | 0.36 | **2.02** | -1.16 | -0.20 | 0.89 | 1.05 |
| PD | 41.08 | 1.70 | **4.18** | -0.86 | -0.40 | 2.63 | 2.79 | 5.40 | 1.11 | **0.68** | -1.46 | -0.03 | 2.34 | 2.41 | 32.14 | 0.12 | **4.20** | -2.31 | 0.13 | 2.04 | 2.17 |

Bolded values denote highest positive outcomes.

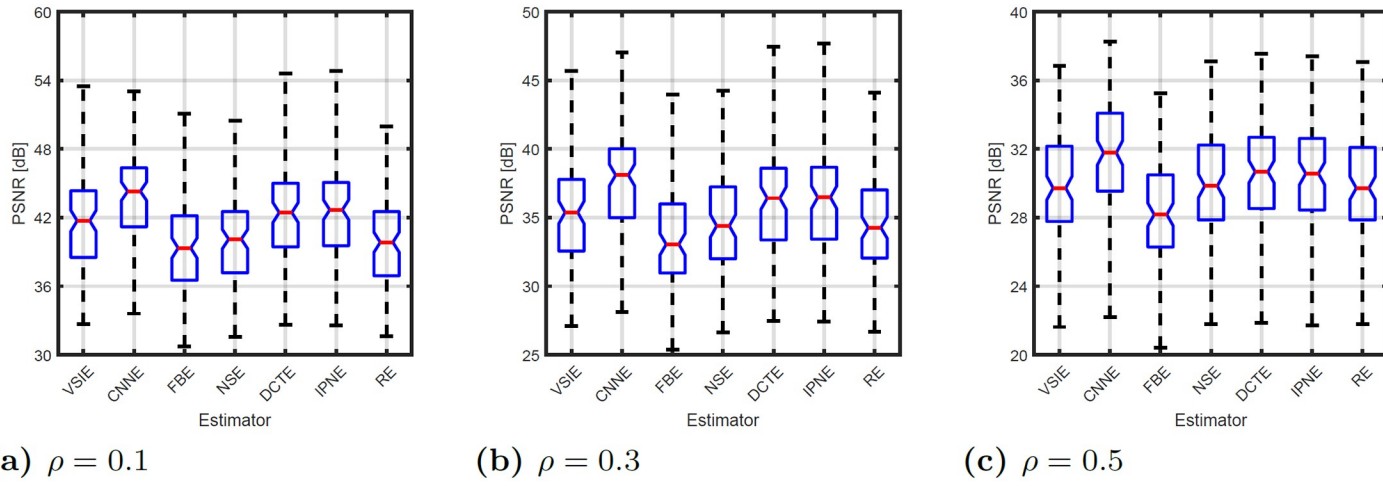

**(a)** $\rho = 0.1$  **(b)** $\rho = 0.3$  **(c)** $\rho = 0.5$

**Fig 3. Impulse replacement performance using the best detector—CNND.**

detector). Although, from a mean ranks perspective, less results show superiority over RE, the general conclusions are rather similar to those drawn from analysis of medians and box plots.

In the sequel to Friedman' tests, Post-hoc testing (proposed by Nemenyi [93]) are conducted, which in our case compared RE to every other algorithm separately. For those tests two new hypothesizes are formulated:

H2:. There is no significant difference in performance of RE and tested algorithm ($p \geq 0.05$).

H3:. Result obtained for tested algorithm is significantly better than obtained using RE ($p < 0.05$).

Results of Post-hoc tests are presented in Table 6 where H2 and H3 hypothesizes are depicted. As comparison of medians and mean ranks may show differences in the average results, the Post-hoc tests state a statistical significance in those differences. Therefore, more H2 results can be found in Table 6, as some averaged results have not been found to be of great effect.

Additionally it is worth noting that:

**Table 5. Mean ranks from Friedman's test.**

| Detector | Estimator | | | | | | | | | | | | | | | | | | | | |
|---|---|---|---|---|---|---|---|---|---|---|---|---|---|---|---|---|---|---|---|---|---|
| | RE | VSIE | CNNE | FBE | NSE | DCTE | IPNE | RE | VSIE | CNNE | FBE | NSE | DCTE | IPNE | RE | VSIE | CNNE | FBE | NSE | DCTE | IPNE |
| | $\rho = 0.1$ | | | | | | | $\rho = 0.3$ | | | | | | | $\rho = 0.5$ | | | | | | |
| ACWD | 2.33 | 4.62 | **6.93** | 1.89 | 2.06 | 4.82 | 5.35 | 5.90 | 4.90 | **6.12** | 1.75 | 4.94 | 2.83 | 1.56 | **6.87** | 4.16 | 5.44 | 2.56 | 5.53 | 2.38 | 1.06 |
| CNND | 2.55 | 4.15 | **6.56** | 1.53 | 2.06 | 5.19 | 5.96 | 2.65 | 3.97 | **6.96** | 1.14 | 2.32 | 5.21 | 5.75 | 3.73 | 3.04 | **7.00** | 1.00 | 3.22 | 5.48 | 4.53 |
| FASTD | 2.30 | 4.11 | **6.82** | 1.43 | 2.41 | 5.07 | 5.86 | 2.70 | 4.34 | **7.00** | 1.14 | 2.45 | 5.24 | 5.13 | 5.65 | 5.25 | **7.00** | 1.62 | 2.74 | 3.66 | 2.08 |
| FFD | 2.37 | 4.19 | **6.93** | 2.06 | 1.76 | 4.95 | 5.74 | 4.83 | 5.63 | **6.79** | 1.43 | 3.64 | 3.50 | 2.18 | **6.73** | 4.37 | 6.05 | 2.27 | 4.85 | 2.52 | 1.21 |
| FPGD | 2.32 | 4.03 | **6.77** | 1.40 | 2.50 | 5.05 | 5.93 | 2.79 | 4.03 | **6.95** | 1.13 | 2.24 | 5.17 | 5.69 | 4.02 | 3.77 | **6.96** | 1.05 | 2.01 | 5.49 | 4.70 |
| GDPD | 2.26 | 4.04 | **6.76** | 1.36 | 2.52 | 5.12 | 5.94 | 2.47 | 3.98 | **6.90** | 1.15 | 2.52 | 5.26 | 5.72 | 3.43 | 3.89 | **6.90** | 1.05 | 2.01 | 5.62 | 5.10 |
| PD | 2.51 | 4.18 | **6.48** | 1.56 | 2.07 | 5.20 | 6.00 | 2.29 | 3.97 | **6.86** | 1.15 | 2.67 | 5.16 | 5.90 | 2.29 | 3.50 | **6.92** | 1.05 | 3.20 | 5.22 | 5.82 |

Highest mean ranks (best results) are emboldened.

**Table 6. Hypothesis for Post-hoc tests.**

| Detector | Estimator | | | | | | | | | | | | | | | | | |
|---|---|---|---|---|---|---|---|---|---|---|---|---|---|---|---|---|---|---|
| | VSIE | CNNE | FBE | NSE | DCTE | IPNE | VSIE | CNNE | FBE | NSE | DCTE | IPNE | VSIE | CNNE | FBE | NSE | DCTE | IPNE |
| | $\rho = 0.1$ | | | | | | $\rho = 0.3$ | | | | | | $\rho = 0.5$ | | | | | |
| ACWD | H3 | H3 | H2 | H2 | H3 | H3 | H2 | H2 | H2 | H2 | H2 | H2 | H2 | H2 | H2 | H2 | H2 | H2 |
| CNND | H3 | H3 | H2 | H2 | H3 | H3 | H3 | H3 | H2 | H2 | H3 | H3 | H2 | H3 | H2 | H2 | H3 | H3 |
| FASTD | H3 | H3 | H2 | H2 | H3 | H3 | H3 | H3 | H2 | H2 | H3 | H3 | H2 | H3 | H2 | H2 | H2 | H2 |
| FFD | H3 | H3 | H2 | H2 | H3 | H3 | H3 | H3 | H2 | H2 | H2 | H2 | H2 | H2 | H2 | H2 | H2 | H2 |
| FPGD | H3 | H3 | H2 | H2 | H3 | H3 | H3 | H3 | H2 | H2 | H3 | H3 | H2 | H3 | H2 | H2 | H3 | H2 |
| GDPD | H3 | H3 | H2 | H2 | H3 | H3 | H3 | H3 | H2 | H2 | H3 | H3 | H2 | H3 | H2 | H2 | H3 | H3 |
| PD | H3 | H3 | H2 | H2 | H3 | H3 | H3 | H3 | H2 | H2 | H3 | H3 | H3 | H3 | H2 | H3 | H3 | H3 |

Decision made with significance level $\alpha = 0.05$.

- the results obtained for all detector-estimator pairs are significantly heterogeneous, in general—Friedman's test rejected H0 for every scenario tested.

- the performance of FBE and NSE is statistically indistinguishable from RE regardless of the noise fraction and detector used.

- CNNE is almost always significantly better than RE with single exception for highest noise fraction and when least accurate detectors (ACWD and FFD) are used.

- the efficiency of VSIE technique is significantly better than that of RE for $\rho < 0.5$ with the exception when ACWD is used as detector.

- other algorithms are more likely better than RE for lower noise fractions, which means that inferiority of RE is less prominent if more impulses occur in the processed image.

- in few test scenarios, when CNNE is used, mean ranks achieved value equal to the number of tested algorithms (see Table 5) which means that CNNE performed best for every processed image.

## An impact of imperfections in detection

The image 48 (MOTOCROSS) from the used dataset, containing a lot of detailed regions, was corrupted with RVIN with intensity $\rho = 0.3$ and selected as test sample. Then in order to evaluate the impact of detection errors on the replacement performance, the CCNE (best estimator) has been paired with every detector (excluding PD), and applied to restore the test image. The evaluation of the result, has been made using Mean Absolute Error (MAE) which is defined for the entire images as:

$$\text{MAE} = \frac{1}{3\theta} \sum_{u=1}^{\mu} \sum_{v=1}^{v} \sum_{q} |o_{u,v}^{q} - x_{u,v}^{q}|. \tag{4}$$

However, we use this index also separately for pixels being TPs, FPs, FNs. The results are presented in circular aim-plots shown in Fig 4 and can be summarized as follows:

- FNs have minimal contribution to MAE in every case, which means that error introduced by omitted impulses has negligible impact on overall denoising performance.

- The efficiency of impulse detection shows noticeable influence in the balance between TPs and FPs contributions. The interesting observation can be made that although the more accurate detectors reduce the magnitude of error for correctly recognized impulses, the

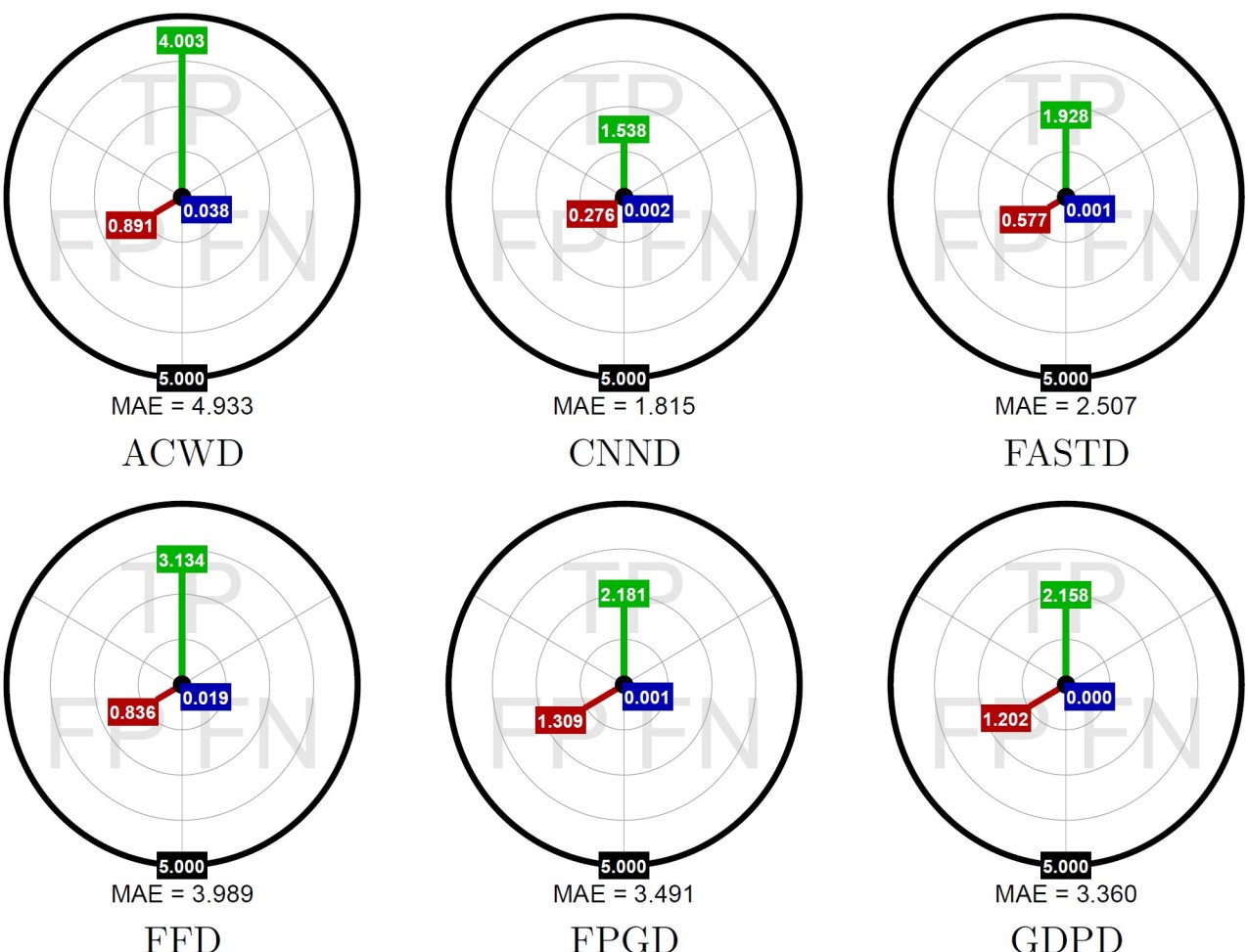

**Fig 4. Aim-plots.** Shows MAE caused by TP, FP, and FN results separately (CNNE was used as the corrupted pixels restoration technique).

portion of MAE cased by FPs is more significant. This can lead to the conclusion, that more accurate impulse replacement has to come at a price of collateral damage made upon small image details, which are incorrectly classified as impulses

## Visual comparison

The visual performance of all tested estimators paired with PD (representing flawless detection) can be observed in Fig 5. Comparison has been once more made using MOTOCROSS image corrupted with RVIN of $\rho = 0.3$.

Results can be concluded as follows:

- without question the CNNE yielded the best performance among tested estimators in both comparisons. The details of the image are well preserved even in heavily textured regions. If ACWD is used some impulses remain (due to insufficient detection accuracy), but textures are well preserved or restored.

- in case of visual comparison, the second best algorithm is DCTE, which also provides very impressive detail and texture preservation.

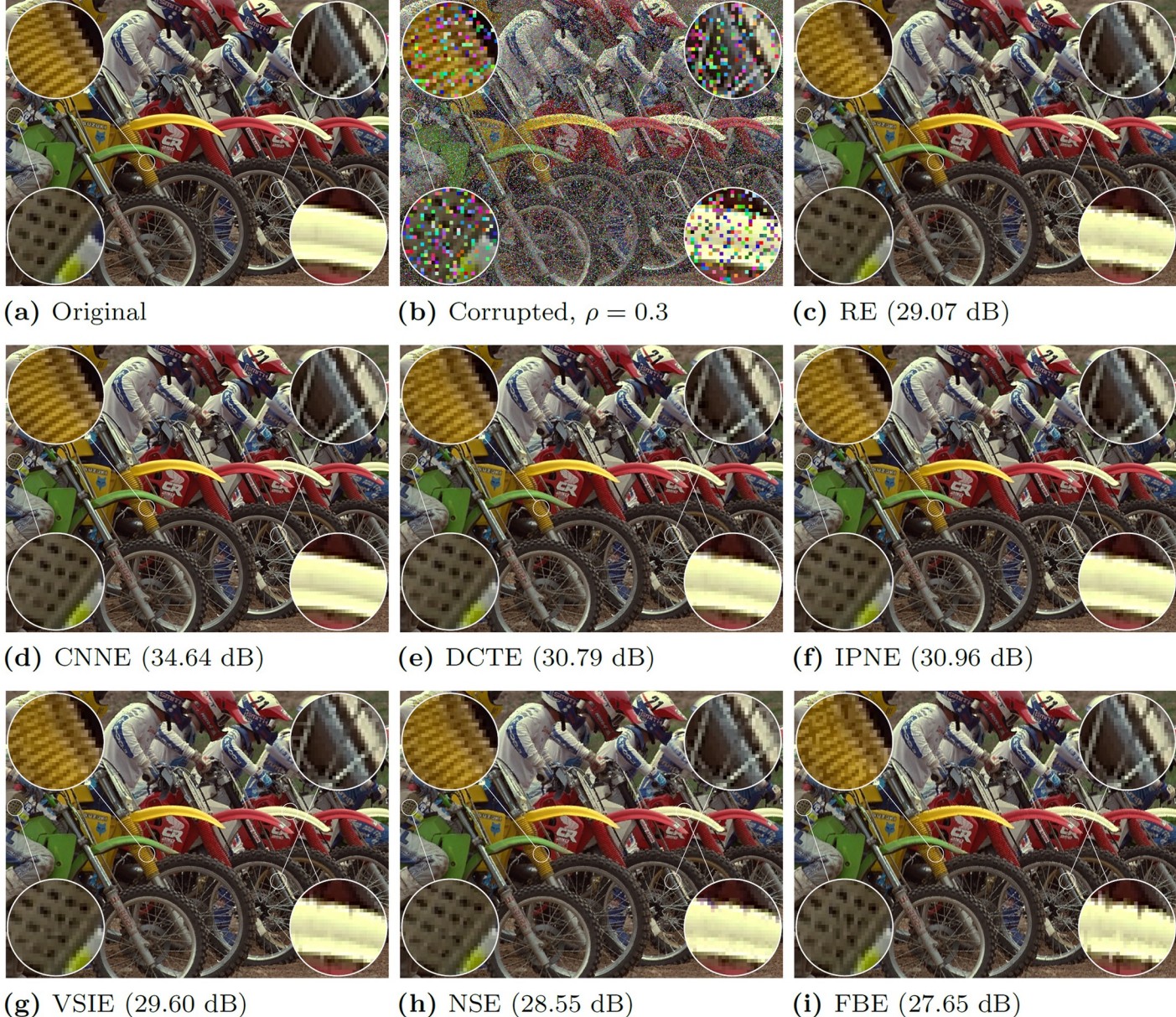

**(a)** Original  **(b)** Corrupted, $\rho = 0.3$  **(c)** RE (29.07 dB)

**(d)** CNNE (34.64 dB)  **(e)** DCTE (30.79 dB)  **(f)** IPNE (30.96 dB)

**(g)** VSIE (29.60 dB)  **(h)** NSE (28.55 dB)  **(i)** FBE (27.65 dB)

**Fig 5. Impulse suppression performance on artificially corrupted image.** MOTOCROSS image ($\rho = 0.3$) using the best detector—CNND and all tested estimators, is presented (PSNR values are provided).

- Other algorithms introduce more errors even if PD is used. The textures are significantly smoothed out and details noticeably deformed.

- as long as the RE algorithm shows inferior efficiency to CNNE and DCTE, it is not the worst algorithm among tested.

- while CNNE just leaves undetected pixels, other algorithms, especially FBE, pollute the surroundings with leftover impulse colors. Therefore their performance is noticeably dependent on detector accuracy.

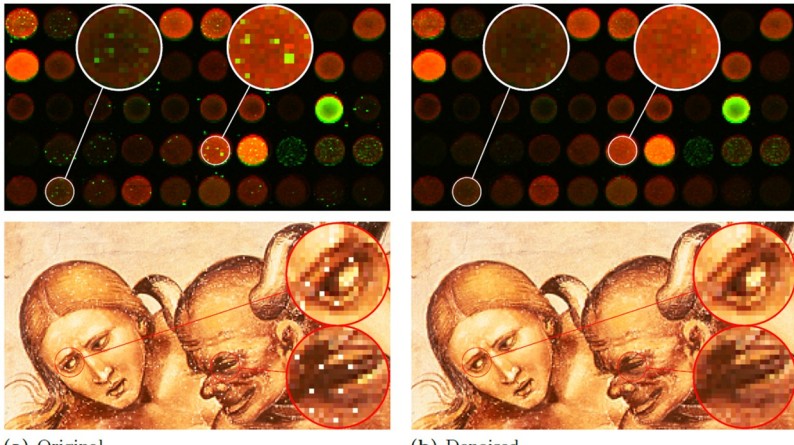

**(a)** Original **(b)** Denoised

**Fig 6. Impulse suppression performance on naturally corrupted images.** Image of cDNA (a) and part of an image of the fresco "The Condemned in Hell" by Luca Signorelli (b) are presented.

The comparison of images in Fig 5 resulted in interesting remark that even though from PSNR point of view DCTE shown statistically inferior numerical efficiency in comparison to IPNE, its actual output is more pleasant visually, as it performs significantly better in edge preservation. Additional examples for different images are provided in S1 File.

The efficiency of the analyzed methods was also evaluated on real images affected by impulsive noise, but for the sake of brevity we presented only the results achieved using the best filter. The first experiment was made on a corrupted cDNA image. This type of images are used for measuring the expression level of large number of genes [94]. The second experiment was performed on a part of the digitized fresco "The Condemned in Hell" by Luca Signorelli. The restoration results are presented in Fig 6 and they confirm good denoising capabilities of the analyzed filter when applied for real noisy images.

## Conclusions

The paper starts with the question: Is large improvement in efficiency of impulsive noise suppression in color images still possible? After extensive testing, performed using numerous impulsive noise detection and interpolation algorithms and thorough analysis of the results, there is no doubt that the answer is indeed positive.

The impulse detection step has a very significant impact on estimation performance because undetected impulses are taken to calculate estimator output, thus introducing estimation errors. Also the occurrence of false positive results lead to a decrease in image quality. However, for low noise contamination fractions, the application of more sophisticated interpolation algorithms than the one used by Reference Estimator (AMF performed not corrupted pixels only) provided noticeable PSNR improvement even if low F1-score detectors are used. If more impulses occur in the image, only a few algorithms outperform the RE, and may yet pore to be inferior, if paired with less accurate detector. Certainly, the best gain in performance is observed if true maps of noise are used for estimation.

Among tested estimators, the best numerical performance is without question achieved by Convolution Neural Network (CNNE), and the second best is Inpaint Nans (IPNE). If visual comparison is performed, CNNE definitely excels other methods but also the Discrete Cosine Transform based technique (DCTE) shows a slightly better performance than IPNE. Other

algorithms, except FBE generally provide improved results when compared to RE, especially if noise fraction is lover than $\rho = 0.5$.

We have shown that combining novel methods of impulse detection and replacement, based on image inpainting techniques and convolutional neural network applied within the framework of switching filters, allows to increase the filtering performance in terms of PSNR by a few decibels, which is a dramatic improvement. The obtained results show clearly that huge gain in performance of various other methods of impulsive and mixed noise suppression can be expected. The application of novel methods of impulse detection and interpolation can be especially advantageous in denoising scenarios, in which the detection of outliers injected into the image by the noise process plays a crucial role. We are positive that the results reported in this paper will promote the development of novel methods of impulsive noise removal, as we have shown that there is still much room for performance gain.

## Supporting information

**S1 File. Additional examples.**
(PDF)

## Author Contributions

**Conceptualization:** Lukasz Malinski, Krystian Radlak, Bogdan Smolka.

**Data curation:** Lukasz Malinski, Krystian Radlak, Bogdan Smolka.

**Formal analysis:** Lukasz Malinski, Krystian Radlak, Bogdan Smolka.

**Funding acquisition:** Lukasz Malinski, Krystian Radlak, Bogdan Smolka.

**Investigation:** Lukasz Malinski, Krystian Radlak, Bogdan Smolka.

**Methodology:** Lukasz Malinski, Krystian Radlak, Bogdan Smolka.

**Project administration:** Lukasz Malinski, Krystian Radlak, Bogdan Smolka.

**Resources:** Lukasz Malinski, Krystian Radlak, Bogdan Smolka.

**Software:** Lukasz Malinski, Krystian Radlak, Bogdan Smolka.

**Supervision:** Lukasz Malinski, Krystian Radlak, Bogdan Smolka.

**Validation:** Lukasz Malinski, Krystian Radlak, Bogdan Smolka.

**Visualization:** Lukasz Malinski, Krystian Radlak, Bogdan Smolka.

**Writing – original draft:** Lukasz Malinski, Krystian Radlak, Bogdan Smolka.

**Writing – review & editing:** Lukasz Malinski, Krystian Radlak, Bogdan Smolka.

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
