## [Decision Letter · Decision Letter 0]

29 Dec 2020

PONE-D-20-23022

Is large improvement in efficiency of impulsive noise removal

in color images still possible?

PLOS ONE

Dear Dr. Malinski,

Thank you for submitting your manuscript to PLOS ONE. After careful consideration, we feel that it has merit but does not fully meet PLOS ONE’s publication criteria as it currently stands. Therefore, we invite you to submit a revised version of the manuscript that addresses the points raised during the review process.

We look forward to receiving your revised manuscript.

Kind regards,

Yiming Tang, Ph.D.

Academic Editor

PLOS ONE

Journal Requirements:

2. In your Data Availability statement, it is unclear why you have selected 'No - some restrictions will apply' . PLOS defines a study's minimal data set as the underlying data used to reach the conclusions drawn in the manuscript and any additional data required to replicate the reported study findings in their entirety. All PLOS journals require that the minimal data set be made fully available. For more information about our data policy, please see http://journals.plos.org/plosone/s/data-availability.

"This work was supported by a research grant 2017/25/B/ST6/02219 from the National

Science Centre, Poland and was also funded by the Silesian University of Technology,

Poland, (grants BK 2020)."

Reviewers' comments:

Reviewer's Responses to Questions

**Comments to the Author**

1. Is the manuscript technically sound, and do the data support the conclusions?

Reviewer #1: Yes

Reviewer #2: Yes

2. Has the statistical analysis been performed appropriately and rigorously? 

Reviewer #1: Yes

Reviewer #2: Yes

3. Have the authors made all data underlying the findings in their manuscript fully available?

Reviewer #1: Yes

Reviewer #2: Yes

4. Is the manuscript presented in an intelligible fashion and written in standard English?

Reviewer #1: Yes

Reviewer #2: Yes

5. Review Comments to the Author

Reviewer #1: Dear editor:

Thank you for inviting me to evaluate the article titled “Is large improvement in efficiency of impulsive noise removal in color images still possible?”. In this paper, the authors evaluate various noisy pixel detection algorithms and replacement techniques and assess their performance by statistical reasoning. Experiments show that detector and estimator based on convolutional neural network achieve the best performance. Finally, the authors draw a conclusion that the improvement in efficiency of impulsive noise suppression in color images is still possible.

The contributions of this work can be summarized as follows:

(1) The introduction of different pixel detection and replacement algorithms.

(2) The implementation of some switching algorithms, including impulse detection stage and impulse replacement stage.

(3) The analysis and evaluation of experiment results and impact of imperfect detection, using tables, figures and statistical reasoning methods.

(4) The visual comparison of performance of tested estimators paired PD.

The article has many strengths. It involves several popular algorithms and explain these algorithms clearly. The whole article consists of many figures and tables, which make the result more intuitive. It has enough comparisons between each algorithm. The evaluations are detailed and thorough. Statistical reasoning methods are used to assess the experiments, which make the analysis clearer and more reliable. Overall, the article is well organized and its presentation is good. However, it also has some weaknesses. First, the style of subtitles in this article is not very clear. Primary and secondary subtitles are distinguished by font size, which is easy to misread. Second, the experiment is only implemented on the synthetic dataset. If some visual comparisons can be made on the real data, the analysis will be more persuasive. Last but not least, the details of the experiment are not clear enough. For example, the dataset and some hyper parameters are not clearly introduced in the training of CNN.

In Section 1, the authors explain the importance of impulsive noise removal and introduce some popular algorithms. Based on these introductions, the authors propose the main purpose of this paper is to find whether a substantial increase in the efficiency of impulsive noise suppression is still possible. However, this part lacks a summary about the structure of this article, which will make the article easier to understand.

In Section 2, the authors show the two stages of switching algorithms, and introduce several algorithms in impulse detections stage and impulse replacement stage. These algorithms are explained in detail, but it is better to do some horizonal comparisons between these methods.

In Section 3, the authors give some common evaluating indicators and present the results of algorithms proposed in Section 2. Statistical reasoning and text descriptions are used to compare their performance. The impact of imperfections in detection is also discussed. The overall structure of this part is very good. The analysis and comparisons are exhaustive. The figures are appropriate too.

In Section 4, the authors make some visual comparisons of all tested estimators paired with PD. There is only one image being comparing in this part. More comparisons on other images should be provided in supplementary materials.

In Section 5, the authors summarize the paper and give the conclusion. This part summarizes well.

In a word, I think this article can be weak accepted.

Additionally, some relevant feature learning based techniques related to image restoration or impulse noise detection should be considered or at least analyzed:

3D Feature Constrained Reconstruction for Low Dose CT Imaging,” IEEE Transactions on Circuits and Systems for Video Technology，28(5), 1232 –1247，2018

“Structure-adaptive Fuzzy Estimation for Random-Valued Impulse Noise Suppression,” IEEE Transactions on Circuits and Systems for Video Technology 28(2), 414 – 427，2018

“Domain Progressive 3D Residual Convolution Network to Improve Low Dose CT Imaging” IEEE, Transactions on Medical Imaging, vol. 38, no. 12, pp. 2903-2913, Dec. 2019

“Discriminative Feature Representation to Improve Projection Data Inconsistency for Low Dose CT Imaging，”IEEE, Transactions on Medical Imaging, vol. 36, no. 12, pp. 2499-2509, 2017.

"Artifact Suppressed Dictionary Learning for Low-dose CT Image Processing,” IEEE, Transaction on Medical Imaging, 33(12), pp.2271-2292,2014

Reviewer #2: This paper investigates noise removal in color images. The authors discuss various impulse noise detection and impulse replacement algorithms.

The paper nicely summarizes the methods and put forwards the results and analyses. I have following suggestions:

1. It will be a better if discussion of various detectors and estimators is put in compact way in a table/figure. This makes the discussion more compelling.

2. The experiments have been performed using dataset containing 100 images with 640x480 pixels resolution.In the experiments, random noise was introduced and then denoising methods were employed. It will be better if authors can include results for real world noisy images where the noise is inherent due to camera/microscope limitations. A figure (Like Fig. 5) showing denoised images using various detection/estimation for a noisy image will improve the results.

6. PLOS authors have the option to publish the peer review history of their article (what does this mean?). If published, this will include your full peer review and any attached files.

Reviewer #1: No

Reviewer #2: **Yes: **Malay Singh

---

## [Author Response · Author response to Decision Letter 0]

10 Feb 2021

We are very grateful for thorough and insightful reviews and we provide following responses:

To Reviewer 1:

1. We are using the available PLOS ONE \\LaTeX{} style, which defines certain section and subsections appearance, and we do not dare to change the official style. Therefore we have no influence on the appearance of subtitles in the preprint version. Hopefully, the paper's final appearance will be improved in the final version of the article, prepared by typesetters.

2. Experiments on real data have been added. Two images naturally contaminated by impulsive noise have been restored using the best combination of detector-estimator and presented in the paper. We provided an example of denoising the cDNA image and an old work of art, which we digitized from a photographic plate using a high quality scanner.

3. Table 2 summarizing the training parameters of the used CNN has been added to the paper. The parameters of the various methods were tuned to achieve the best possible results. All the images (original, noisy and restored) have been made available on KAGGLE, so that the research community can use them to compare their results with the output of the filters we have chosen for comparisons. You are right that the description of all the parameters would be advantageous, however a full list of their values, applied for each image of the dataset and noise intensity would be very lengthy and many huge tables would be required. Therefore we did not attempt to provide the detailed analysis of all used parameters of the various filtering techniques and their exact values. Instead we enabled the readers to download the final, optimal filtering result. If desired, the same filtering outputs can be obtained using the optimized combination of the filters' parameters described in the respective papers.

4. The brief summary has been added and the end of the introduction Section. Thank you for pointing out its absence, as it really improved the readability of the paper.

5. If we have understood it correctly, the vertical comparison means evaluating the performance of a switching filter as a whole and the horizontal comparison means comparing detection and replacement algorithms separately. If our reasoning is correct, these kind of horizontal comparisons are already included in the paper. The separate comparison between detectors is presented in Tab. 1, where Accuracy and F1 scores are shown. For the impulse replacement part, the horizontal comparison is reflected by the evaluation performed using the Perfect Detector (PD), for which the impact of the detection performance is eliminated.

6. Additional 6 thoroughly selected examples have been added to the paper as support information. We made additional comments on the efficiency of the evaluated filters showing their advantages and weaknesses depending on the image structure type and noise contamination intensity. We hope that the additional examples and also the included aim-plots will enable the reader to to get an insight into the filtering properties of the methods taken for comparisons.

7. We have included a short description of suggested denoising framework in the Introduction.

To Reviewer 2:

1. Experiments on real data have been added. Two images naturally contaminated by impulsive noise have been restored using the best combination of detector-estimator and presented in the paper. We provided an example of denoising the cDNA image and an old work of art, which we digitized from a photographic plate using a high quality scanner.

2. We summarized all descriptions of both detection and estimation algorithms in one compact Table. We hope that such a modification improves the readability of the paper.

---

## [Decision Letter · Decision Letter 1]

31 May 2021

Is large improvement in efficiency of impulsive noise removal

in color images still possible?

PONE-D-20-23022R1

Dear Dr. Malinski,

We’re pleased to inform you that your manuscript has been judged scientifically suitable for publication and will be formally accepted for publication once it meets all outstanding technical requirements.

Kind regards,

Yiming Tang, Ph.D.

Academic Editor

PLOS ONE

Additional Editor Comments (optional):

Reviewers' comments:

Reviewer's Responses to Questions

**Comments to the Author**

1. If the authors have adequately addressed your comments raised in a previous round of review and you feel that this manuscript is now acceptable for publication, you may indicate that here to bypass the “Comments to the Author” section, enter your conflict of interest statement in the “Confidential to Editor” section, and submit your "Accept" recommendation.

Reviewer #2: All comments have been addressed

Reviewer #3: All comments have been addressed

2. Is the manuscript technically sound, and do the data support the conclusions?

Reviewer #2: Yes

Reviewer #3: Yes

3. Has the statistical analysis been performed appropriately and rigorously? 

Reviewer #2: Yes

Reviewer #3: Yes

4. Have the authors made all data underlying the findings in their manuscript fully available?

Reviewer #2: Yes

Reviewer #3: Yes

5. Is the manuscript presented in an intelligible fashion and written in standard English?

Reviewer #2: Yes

Reviewer #3: Yes

6. Review Comments to the Author

Reviewer #2: The authors have provide reasonable response to my queries. They have added a compact Table and example images for selected noise removal method.

Reviewer #3: I think the manuscript in its present condition meets the standards for publication.Therefore, I have no further questions.

7. PLOS authors have the option to publish the peer review history of their article (what does this mean?). If published, this will include your full peer review and any attached files.

Reviewer #2: **Yes: **Malay Singh

Reviewer #3: No

---

## [Editor Report · Acceptance letter]

17 Jun 2021

PONE-D-20-23022R1 

Is large improvement in efficiency of impulsive noise removal in color images still possible? 

Dear Dr. Malinski:

I'm pleased to inform you that your manuscript has been deemed suitable for publication in PLOS ONE. Congratulations! Your manuscript is now with our production department. 

Kind regards, 

on behalf of

Professor Yiming Tang 

Academic Editor

PLOS ONE